# Impact of gaseous ozone treatment of fish carcasses (*Gadus morhua*) on the microbiological load and their quality

Piotr Antos[1]*, Tomasz Piechowiak[2], Maciej Balawejder[2], Karolina Kowalczyk[1], Krzysztof Tereszkiewicz[1]

1 Department of Computer Engineering in Management, Faculty of Management, Rzeszow University of Technology, Rzeszów, Poland, 2 Department of Chemistry and Food Toxicology, Faculty of Technology and Life Sciences, University of Rzeszow, Rzeszów, Poland

* p.antos@prz.edu.pl

## Abstract

The improvement of the microbial quality of food products is one of the most important aspects of the world economy with a significant impact on the health of consumers. Due to the content of lipids, water, and proteins, meat is highly susceptible to degradation mediated by enzymes or microorganisms. Among the wide variety of animal-derived consumption products, fish is highly perishable and characterized by a short shelf life. To improve safety and microbial quality during storage, fish food can be treated with ozone, which is a highly potent antimicrobial agent. In this study, the effect of ozone treatment on the total microorganism content and the quality of fish food was investigated with respect to the content of lipids and the volatile faction responsible for the characteristic odour. The content of microorganisms in fish food subjected to an atmosphere enriched with ozone was reduced. For example, for the control sample with $1.8 \times 10^3$ cfu, which was exposed to 20 ppm ozone in atmosphere during the exposition time in the range 10–30 min, a microbial load was reduced to a level between $1.6 \times 10^3$ and $1.2 \times 10^3$ cfu. The observed reduction levels indicated that such an ozone treatment procedure can be declared as an viable option for improvement of microbial safety of fish. However, the presence of ozone caused qualitative changes of chemical parameters of the fish matrix; in particular, oxidation of lipids occurred. The extent of those changes increased with the extension of the exposition time..

## Introduction

Among many food products, meat is probably one of the most important commodities. Especially in the case of fish, it is considered a highly perishable matrix that requires proper food safety techniques to maintain its quality and extend its shelf life. The changes that occur within this matrix after harvesting are mediated by bacterial growth, enzymatic, and chemical activity.

**Data availability statement:** All relevant data are within the paper and its Supporting Information files.

**Funding:** The author(s) received no specific funding for this work.

**Competing interests:** The authors have declared that no competing interests exist.

The microbial-mediated oxidation of proteins can also lead to changes in texture, causing loss of firmness and increase in water holding capacity. This can result in a mushy texture and reduced overall quality of the fish, as reported by [1–3]. Other authors indicated that the flavour of fish meat may be modified by increasing the number of volatile compounds generated during microbiologically mediated oxidative degradation of the fish matrix, resulting in the generation of compounds such as aldehydes, alcohols, and terpenes that may be generated as a result of lipid degradation [4].

The oldest methods used for the preservation of fish are the use of salt and smoking. In recent years, modern preservation methods, such as freezing, vacuum packaging, MAP, and HPP, have gained popularity due to their effectiveness in maintaining fish quality. These methods are effective in slowing the growth of microorganisms and enzymatic activity, as well as maintaining the texture and flavour of fish [5–10].

However, there is potential for another approach to food preservation. Ozone treatment not only slows growth, but also inactivates bacteria, viruses, and parasites that are commonly found in fish and seafood. This method of disinfection has been found to be more effective than traditional methods such as chlorine or UV irradiation [11]. For example, ozone has been found to be effective in reducing microbial growth and extending the shelf life of food products, including chicken meat [12].

Ozone in combination with ice slurry has also been found to improve the quality of fish and seafood [13]. Ozone, especially as a water solution treatment, can help reduce fishy odor and improve the microbial safety of fish and seafood. This can cause an increase in consumer acceptability and demand for fish products [14]. Therefore, ozone was found to be a suitable agent for use as part of protection and preservation treatment methods for food products of plant and animal origin [15,16].

Although there is a great deal of data on ozone treatment in regard to microbial safety, there is a lack of data on the complex effects of the interaction of ozone with fish products. In particular, there is a literature gap on the effect of ozonation on the chemical composition of fish as well as other determinants of fish quality.

The main purpose of the study was to investigate the potential of ozone treatment to improve microbial safety and determine the impact of ozonation on other quality characteristics of the fish product. Since the microbial based oxidation of lipids can be hindered by using ozone, other questions were open, i.e., if the antimicrobial factor could cause some qualitative changes in other parameters of the fish matrix, such as antioxidant activity, lipid content, or odour. To provide an answer for those questions, several tests of the quality of fish subjected to ozonation were performed, including measurement of antioxidant activity, determination of oxidation products of lipid degradation, determination of fatty acid profile, and SPME methods for determination of changes in the volatile faction responsible for fish odour.

## Materials and methods

### Acquisition of food products

The (*Gadus morhua*) fish carcass samples (in total 24 filets) were obtained from a Polish company and transferred from the fisheries to the laboratory in cold storage, which provided the security of the cold chain. Since the fish matrix consisted of dead

organisms, no experiments on living organisms were conducted. Therefore, no agreement of the Institutional Animal Care and Use Committee on research on living subjects was needed. The samples were then divided under sterile conditions.

## Ozonation procedure

In the ozonation procedure, fish samples were exposed for 10, 20, or 30 minutes to ozone enriched atmosphere with a concentration of 20 ppm. For this purpose, a Korona AIR-WATER ozone generator (Piotrków Trybunalski, Poland) was used; the ozone concentration was verified by the Ozone Analyzer UV-106MH (Ozone Solutions, Hull, IA, USA).

## Microbiological assessment

The samples before and after ozonation were stored in a freezer at −19 ° C. In total, 24 fish filets were used: 3 as a control sample and 18 as research samples. Then, the microbiological testing was conducted by the GBA Polska company. The investigation was carried out according to the Polish standard, which specifies a horizontal method to determine the number of microorganisms capable of growth and colony formation in solid media after incubation under aerobic conditions at 30 ° C: [17].

## Chemical research

To determine the quality of ozonated fish, several tests were performed, including measurements of antioxidant activity, the level of malondialdehyde, which is a metabolite of lipid oxidation, the determination of the fatty acid profile during the ozonation procedure and the changes in odour measured with the SPME method.

a)  Antioxidant activity

According to the procedure used in a previous research by [18], in order to determine the antioxidant activity, 5 g of tissue were homogenized with 10 mL of a 75% methanol solution. The homogenate was then centrifuged at 10,000g for 30 min, after which the supernatant was used for analysis. Antioxidant activity was determined using ABTS radicals according to the protocol presented by [18,19].

b)  Level of malondialdehyde

The level of malondialdehyde, which is a characteristic metabolite of lipid oxidation, was determined by the following procedure: a 100 mg tissue sample was homogenized with 1.5 ml of 0.5% thiobarbituric acid in 10% trichloroacetic acid. The homogenate was then heated in a boiling water bath for 20 min, then cooled and centrifuged at 10,000 for 30 min. The absorbance of the supernatant was measured at 532 nm. The molar absorption coefficient of the MDA-TBA adduct of 155 mM/cm was used for the calculations [20].

c)  Fatty acid profile

To determine the fatty acid profile, the sample was prepared by extracting the fat from the tissue with methylene chloride: methanol, saponifying the fat with a sodium hydroxide solution (in methanol), and esterifying the separated fatty acid salts with boron trifluoride (in methanol). Fatty acid ester analysis was performed using a Varian GC-450 gas chromatograph coupled with a 4000/240 MS mass spectrometer. The separation conditions are presented in the paper [20].

d)  SPME analysis of volatile faction.

To determine the profile of volatile substances contained in fish, a 5 g sample was placed in a bottle with a rubber septum and thermally stabilised at 40 ° C for 15 min, to achieve maximum vapour pressure of volatile substances. Subsequently, the analytes were separated using a 100 μm polydimethylsiloxane (PDMS) fibre manufactured by Supelco Ltd. (Bellefonte, PA, USA) by exposition of the fiber for 30 min.

e) Chromatographic analysis

The composition of the desorbed compounds was examined using the Varian 450 GC gas chromatograph with the Varian 240 MS mass detector (GC-MS), according to the method proposed by [21].

## Statistical analysis

The obtained results were analysed using STATISTICA 13.1 software (StatSoft, Palo Alto, CA, USA). One-way analysis of variance (ANOVA) was applied (α = 0.05) to show the differences between observed results.

## Results and discussion

### Determination of microbial load

The ozone dose can be modified by changing in the exposition time at the same ozone concentration or by changing the ozone concentration at the same exposition time In this study, the ozone dose was controlled by modifying the exposition time at constant ozone concentrations. To analyze the efficacy of ozonation treatment and examine the complex effects accompanying ozonation, the fish samples were exposed to ozone within three different time intervals at a concentration of 20 ppm. This concentration level was found to be sufficient to reduce the microbial load in previous studies, e.g., [15].

The results of microbiological tests indicated that a reduction in the microbial load occurred and was above the observed error levels; hence, as such, the procedure matched its objectives. For example, the microbial load of the control sample amounted 1.8 x $10^3$ colony forming units (cfu), was reduced to levels oscillating between 1.6 x $10^3$ and 1.2 x $10^3$ for the exposition time changed between 10 and 30 min. The SD for the observed means of microorganism count before and after ozone exposure of fish was +/- 15% e.g., between 2.7*$10^2$ and 1.75*$10^2$

The observed reduction levels indicated that under the condition used, the ozone treatment can improve the microbial safety of fish

### Antioxidant potential of ozonated fish

Although ozone is effective in reducing the population of microorganisms present on the surface of the fish, it may happen that gaseous ozone could affect other vital parameters of the fish meat. An important parameter of fish quality that could be affected as a result of exposure of fish tissue to the ozone enriched atmosphere is the antioxidant potential. This, in turn, may be analysed with the ABTS assay (see Fig 1), which is frequently used to determine the antioxidant potential of food matrixes [18].

The Anova method was used for statistical analysis. The values obtained, which are marked in Fig 1 with the same letters, are not statistically significantly different at the significance level of 0.05.

The analysis of the results showed that the labile chemical compounds in the carcasses of fish remained intact by the action of ozone, even under conditions characterized by high oxidizing potential. The ozone oxidation potential of E0 = 2.07 V did not negatively affect the chemical stability of the antioxidant components present in the carcasses of fish.

### Metabolites of lipids oxidation

The analysis of the impact of the ozone enriched atmosphere on the matrix properties revealed that an increase of the exposure time of fish to the ozone enriched atmosphere has impact on the average level of malondialdehyde. The mean level of this compound was higher for the longer exposure to the ozone factor and was growing constantly after 10, 20, and 30 minutes of the exposition time as presented in Fig 2. The Anova method was used for the statistical analysis; the values obtained are presented in Fig 2 in the similar way as in Fig 1 Significant differences between the results are indicated by different letters; significant differences are defined by the criterion α < 0.05.

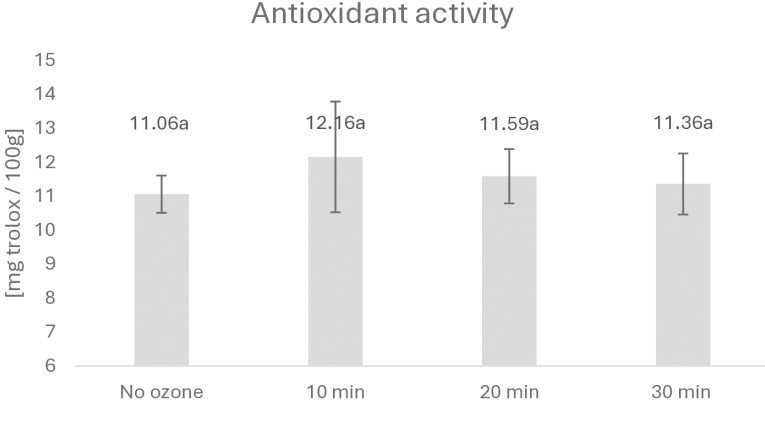

**Fig 1. Illustration of the changes in antioxidant activity during the ozonation of fish tissue.**

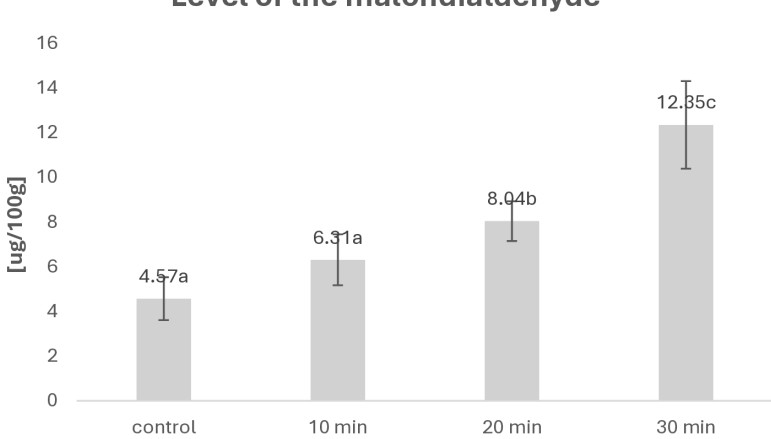

**Fig 2. Level of malondialdehyde generated during peroxidation of lipids.**

A clear dose-response relationship was found for the level of oxidation of the fish fat fraction with the resulting R parameter 0.97. The increase in malondialdehyde is a clear sign of the interaction of ozone with lipids present in the fish matrix [22–24].

This in turn was analysed by the determination of changes in fatty acid profile during the ozonation procedure. The observed compounds and their changes are summarized in Table 1.

The analysis of the fatty acid profile revealed various effects of ozone. The highest increase in the concentration of fatty acids was observed for 11-octadecadienoic acid, while the highest decrease in the compound concentration was observed for 5,8,11,14-eicosatetraenoic acid and for 5,8,11,14,17-eicosapentaenoic acid, which both had a number of unsaturated bonds that could be affected by ozone.

As a result of the ozonation process, aldehydes, ketones, and carboxylic acids can be formed, including 11-octadecadienoic acid, which is correlated with an increase in the level of malondialdehyde. This observation is crucial for the ultimate goal of the investigation, since it revealed an alternate route of lipid oxidation that was hoped to be avoided as a result of the deactivation of microorganisms.

**Table 1. Fatty acid profile.**

| | % of concentration (n = 3) | | | |
|---|---|---|---|---|
| | 0 | 10 min | 20 min | 30 min |
| Tridecanoic acid, 12-methyl- | 1.74 | 2.44 | 2.58 | 1.77 |
| Tetradecanoic acid, 13-methyl- | 0.73 | 1.05 | 1.25 | 1.14 |
| Pentadecanoic acid, 14-methyl- | 14.53 | 19.64 | 19.51 | 15.37 |
| 9-cis, 11-trans-octadecadienoic acid | 1.54 | 2.27 | 1.39 | 1.42 |
| 11-octadecadienoic acid | 12.59 | 18.54 | 30.82 | 42.64 |
| Heptadecanoic acid, 14-methyl- | 7.75 | 10.52 | 9.44 | 1.42 |
| 5,8,11,14-eicosatetraenoic acid | 41.86 | 26.02 | 20.55 | 20.78 |
| 5,8,11,14,17-eicosapentaenoic acid | 19.24 | 19.49 | 14.43 | 15.43 |

## Analysis of volatile faction

Next, the SPME analysis revealed that ozonation caused a reduction in the level of some smell-responsible compounds including 1,3,6-octatriene, decanal and decenal. On the other hand, some compounds responsible for smell were found to be generated as a result of oxidation based transformation, which can be observed for example pentadecanal (see Table 2).

## Conclusions

The use of the ozone procedure resulted in a statistically significant reduction of the microbial load expressed as the amount of colony forming units on the fish matrix This indicates, a potential for extension of shelf life of fish products. However, various qualitative changes occurred as a complex effect within the ozonated matrix. Although ozonation of fish hinders oxidation mediated by microorganisms, it can also lead to the alternative pathway of lipid oxidation, which can result in a reduction in their nutritional value. This was verified by the analysis of fatty acids compounds. Furthermore, the flavour of fish could also be affected as a result of the ozonation of the volatile fraction. On the basis of the performed analysis, it was concluded that most of the compounds responsible for fish odour underwent a reduction. In general, the use of ozone to improve fish-derived food characteristics, such as microbial safety or shelf life, should be carried out at the shortest possible exposure time, for example, not longer than 10 min, due to side effects, such as a reduction in the lipid fraction of the food matrix.

**Table 2. Peak share of the volatile faction identified with the SPME method.**

| Retention Time [min] | Peak share in the chromatogram [%] | | | | | | | | compound |
|---|---|---|---|---|---|---|---|---|---|
| | 0 | | 10 min | | 20 min | | 30 min | | |
| | area | share [%] | area | share [%] | area | share [%] | area | share [%] | |
| 9.11 | 22089 | 4.61 | 74814 | 3.68 | 137968 | 5.47 | 29150 | 2.12 | Sabinene |
| 9.49 | 108218 | 22.59 | 416351 | 20.51 | 534006 | 21.17 | 194576 | 14.12 | 3-Carene |
| 9.88 | 104711 | 21.86 | 392190 | 19.32 | 585207 | 23.20 | 186608 | 13.54 | DL-Limonen |
| 10.27 | 24256 | 5.06 | 94097 | 4.63 | 142089 | 5.63 | 42164 | 3.06 | trans-beta-Ocimene |
| 11.03 | 75236 | 15.71 | 235750 | 11.61 | 317220 | 12.58 | 131467 | 9.54 | Terpinolene |
| 11.29 | 83639 | 17.46 | 395649 | 19.49 | 370193 | 14.68 | 377065 | 27.37 | Decanal |
| 13.32 | 45163 | 9.43 | 96111 | 4.73 | 123178 | 4.88 | 83754 | 6.08 | Decenal |
| 14.45 | 15670 | 3.27 | 325353 | 16.02 | 312517 | 12.39 | 333019 | 24.17 | Pentadecanal |
| | Σ = 478982 | | Σ = 2030315 | | Σ = 2522378 | | Σ = 1377803 | | |

## Supporting information

**S1 Chemical results Fig 1. Antioxidant activity.**
(S1 Antioxidant activity Fig 1.TXT)

**S2 Chemical results Fig 2. Level of malondialdehyde.**
(S2 Level of malondialdehyde Fig 2.TXT)

**S3 Fig. Microbiology results.**
(S3 Microbiology.TXT)

## Author contributions

**Conceptualization:** Piotr Antos, Krzysztof Tereszkiewicz.

**Formal analysis:** Piotr Antos.

**Investigation:** Piotr Antos, Tomasz Piechowiak, Maciej Balawejder.

**Methodology:** Maciej Balawejder.

**Project administration:** Piotr Antos.

**Resources:** Piotr Antos, Tomasz Piechowiak, Maciej Balawejder.

**Supervision:** Piotr Antos.

**Visualization:** Maciej Balawejder.

**Writing – original draft:** Piotr Antos.

**Writing – review & editing:** Piotr Antos, Karolina Kowalczyk.

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
