## [Decision Letter · Decision Letter 0]

25 Nov 2024

PONE-D-24-48297Impact of gaseous ozone treatment of fish carcasses (Gadus morhua) on the microbiological load and their quality.PLOS ONE

Dear Dr. Antos,

Thank you for submitting your manuscript to PLOS ONE. After careful consideration, we feel that it has merit but does not fully meet PLOS ONE’s publication criteria as it currently stands. Therefore, we invite you to submit a revised version of the manuscript that addresses the points raised during the review process.

We look forward to receiving your revised manuscript.

Kind regards,

José Miguel Alvarez-Suarez

Academic Editor

PLOS ONE

2. We note that your Data Availability Statement is currently as follows: [All relevant data are within the manuscript and its supporting information files.] Please confirm at this time whether or not your submission contains all raw data required to replicate the results of your study. Authors must share the “minimal data set” for their submission. PLOS defines the minimal data set to consist of the data required to replicate all study findings reported in the article, as well as related metadata and methods (https://journals.plos.org/plosone/s/data-availability#loc-minimal-data-set-definition). For example, authors should submit the following data: - The values behind the means, standard deviations and other measures reported; - The values used to build graphs; - The points extracted from images for analysis. Authors do not need to submit their entire data set if only a portion of the data was used in the reported study. If your submission does not contain these data, please either upload them as Supporting Information files or deposit them to a stable, public repository and provide us with the relevant URLs, DOIs, or accession numbers. For a list of recommended repositories, please see https://journals.plos.org/plosone/s/recommended-repositories. If there are ethical or legal restrictions on sharing a de-identified data set, please explain them in detail (e.g., data contain potentially sensitive information, data are owned by a third-party organization, etc.) and who has imposed them (e.g., an ethics committee). Please also provide contact information for a data access committee, ethics committee, or other institutional body to which data requests may be sent. If data are owned by a third party, please indicate how others may request data access.

Additional Editor Comments (if provided):

Reviewers' comments:

Reviewer's Responses to Questions

**Comments to the Author**

1. Is the manuscript technically sound, and do the data support the conclusions?

Reviewer #1: No

Reviewer #2: Yes

2. Has the statistical analysis been performed appropriately and rigorously? 

Reviewer #1: No

Reviewer #2: I Don't Know

3. Have the authors made all data underlying the findings in their manuscript fully available?

Reviewer #1: No

Reviewer #2: Yes

4. Is the manuscript presented in an intelligible fashion and written in standard English?

Reviewer #1: Yes

Reviewer #2: Yes

5. Review Comments to the Author

Reviewer #1: Summary and General Comments

The manuscript explores the effect of ozone treatment on microbial load and chemical quality parameters of fish carcasses. The study's aim—to assess the balance between microbial safety improvement and chemical stability of fish quality under ozone exposure—has potential relevance in food preservation research. However, the study suffers from several critical flaws that affect the validity and impact of its findings. Specifically, the manuscript lacks rigor in methodological design, clarity in data interpretation, and provides insufficient discussion on the broader implications of the results.

Major Concerns

Experimental Design and Controls:

The study's methodological approach to control sample treatment lacks detail, specifically regarding storage conditions and any baseline tests performed. Further, the exposure times to ozone (10, 20, and 30 minutes) do not account for varying ozone concentrations, leaving questions about the applicability of these results under different conditions.

The authors do not describe the basis for selecting 20 ppm as the ozone concentration or how this relates to commonly accepted or regulatory levels for ozone exposure in food.

Data Analysis and Statistical Interpretation:

The statistical analysis is inadequately explained. The manuscript lacks an account of whether differences in microbial counts and quality parameters between time points are statistically significant. For instance, the authors state reductions in microbial load but do not clarify if these reductions are meaningful or consistent across replicates.

There is also insufficient discussion on the standard deviations or confidence intervals, which would strengthen the reliability of the results presented.

Impact on Quality Parameters and Lack of Comprehensive Discussion:

The manuscript mentions the effect of ozone on lipid oxidation by tracking malondialdehyde levels and changes in the fatty acid profile, yet the discussion is brief and fails to integrate these results into a broader context. For example, the implications of increased oxidation on fish quality and nutritional value are not thoroughly discussed.

The volatile profile analysis lacks depth, particularly regarding how the detected compounds impact consumer perception and shelf life. The authors should compare the sensory characteristics of treated fish to typical quality standards.

Literature Review and Contextualization:

The literature review is somewhat outdated, and the authors miss recent advancements in non-thermal preservation methods that could provide context for their findings. There is also insufficient discussion of similar studies on ozone treatment of fish or other perishable products.

Language and Clarity Issues:

The manuscript requires extensive language revision to improve clarity and readability. Many sections contain ambiguous statements, grammar errors, and redundancies, which hinder the reader’s ability to understand the study's findings and implications.

Minor Concerns

Figures and Tables:

Some figures, such as those showing the malondialdehyde levels and microbial load, are not labeled clearly, and it is difficult to interpret their significance without adequate legends and discussion.

Table 2, which presents the volatile fraction, lacks proper labeling of columns and rows, making it challenging to understand the changes in volatile compounds.

Reviewer #2: This manuscript is about the Impact of gaseous ozone treatment of fish carcasses (Gadus morhua) on the microbiological load and their quality. My recommendation on submitting the manuscript to PLOS ONE is acceptation after major revision. Please see detailed comments and questions as below

-In Title, Scientific name (Gadus morhua) should be written in italic format.

-In the abstract, it is better to mention the values of the results obtained.

- At the end of abstract, the keywords starts with first capital letter.

-The introduction section is long and needs to be rewritten.

-Due to the effect of ozone on food fat and the creation of an undesirable taste, It is better that sensory evaluation performed on the samples.

-Comparison of results with those of other similar studies, as well as discussion in the results section, is weak.

-Please insert significant letters above the columns in figures.

6. PLOS authors have the option to publish the peer review history of their article (what does this mean? ). If published, this will include your full peer review and any attached files.

**Do you want your identity to be public for this peer review?** For information about this choice, including consent withdrawal, please see our Privacy Policy .

Reviewer #1: No

Reviewer #2: No

---

## [Author Response · Author response to Decision Letter 1]

22 Jan 2025

Reviewer #1: Summary and General Comments

The manuscript explores the effect of ozone treatment on microbial load and chemical quality parameters of fish carcasses. The study's aim—to assess the balance between microbial safety improvement and chemical stability of fish quality under ozone exposure—has potential relevance in food preservation research. However, the study suffers from several critical flaws that affect the validity and impact of its findings. Specifically, the manuscript lacks rigor in methodological design, clarity in data interpretation, and provides insufficient discussion on the broader implications of the results.

Major Concerns

Experimental Design and Controls:

The study's methodological approach to control sample treatment lacks detail, specifically regarding storage conditions and any baseline tests performed. Further, the exposure times to ozone (10, 20, and 30 minutes) do not account for varying ozone concentrations, leaving questions about the applicability of these results under different conditions.

The authors do not describe the basis for selecting 20 ppm as the ozone concentration or how this relates to commonly accepted or regulatory levels for ozone exposure in food.

Action taken:

In section “determination of microbial load” a comment was added:

The ozone dose can be modified by changing in the exposition time at the same ozone concentration or by changing the ozone concentration at the same exposition time In this study, the ozone dose was controlled by modifying the exposition time at constant ozone concentrations. To analyze the efficacy of ozonation treatment and examine the complex effects accompanying ozonation, the fish samples were exposed to ozone within three different time intervals at a concentration of 20 ppm. This concentration level was found to be sufficient to reduce the microbial load in previous studies, e.g, Antos et al., 2020.

Data Analysis and Statistical Interpretation:

The statistical analysis is inadequately explained. The manuscript lacks an account of whether differences in microbial counts and quality parameters between time points are statistically significant. For instance, the authors state reductions in microbial load but do not clarify if these reductions are meaningful or consistent across replicates.

There is also insufficient discussion on the standard deviations or confidence intervals, which would strengthen the reliability of the results presented.

Action taken:

Additional section statistical analysis was added:

“The obtained results were analysed using STATISTICA 13.1 software (StatSoft, Palo Alto, CA, USA). One-way analysis of variance (ANOVA) was applied (α = 0.05) to show the differences between observed results.

In section “Determination of microbial load” an information on the SD values for the determined microbial loads was added.

“The SD for the observed means of microorganism count before and after ozone exposure of fish was +/- 15% e.g., between 2.7*102 and 1.75*102“

In section “Antioxidant potential of ozonated fish” a comment was added:

The Anova method was used for statistical analysis. The values obtained, which are marked in Fig. 1 with the same letters, are not statistically significantly different at the significance level of 0.05.

In section “Metabolites of lipids oxidation” comment have been added:

The Anova method was used for the statistical analysis; the values obtained are presented in Fig. 2 in the similar way as in Fig. 1 Significant differences between the results are indicated by different letters; significant differences are defined by the criterion α < 0.05.

Also the relevant figures were improved by addition of significance letters.

Impact on Quality Parameters and Lack of Comprehensive Discussion: The manuscript mentions the effect of ozone on lipid oxidation by tracking malondialdehyde levels and changes in the fatty acid profile, yet the discussion is brief and fails to integrate these results into a broader context. For example, the implications of increased oxidation on fish quality and nutritional value are not thoroughly discussed.

Action taken: in the introduction section a comment was added:

Although there is a great deal of data on ozone treatment in regard to microbial safety, there is a lack of data on the complex effects of the interaction of ozone with fish products. In particular, there is a literature gap on the effect of ozonation on the chemical composition of fish as well as other determinants of fish quality.

The volatile profile analysis lacks depth, particularly regarding how the detected compounds impact consumer perception and shelf life.

Action taken:

In section “Sensory analysis” a proper comment was added:

A preliminary sensory analysis of the fish tissue performed without heat treatment indicated the presence of foreign odour notes, which are typical of carbonyl products of fatty acid decomposition. Furthermore, in the volatile fraction, a representative of these compounds was identified, i.e., reported above pentadecanal (see Table 2). Due to its high volatility, this compound will not have a significant impact on the perception of consumers of heat treated products.

In section “Conclusion” a proper comment was added:

The use of the ozone procedure resulted in a statistically significant reduction of the microbial load expressed as the amount of colony forming units on the fish matrix This indicates, a potential for extension of shelf life of fish products.

The authors should compare the sensory characteristics of treated fish to typical quality standards.

The response for this comment is given above.

Literature Review and Contextualization:

The literature review is somewhat outdated, and the authors miss recent advancements in non-thermal preservation methods that could provide context for their findings.

Action taken:

The literature was updated in the introduction section by addition some recent positions (Gouvea et al., 2023; Nie et al., 2022)

There is also insufficient discussion of similar studies on ozone treatment of fish or other perishable products.

Response:

There is a literature gap on the subject of complex effects of ozonation treatment. In particular, only microbial load effect is investigated, while effects like changes in fatty acid profile or volatile faction composition were not the subject of any of known literature positions available.

Action taken:

In the introduction section a proper comment was added:

“Although there is a great deal of data on ozone treatment in regard to microbial safety, there is a lack of data on the complex effects of the interaction of ozone with fish products. In particular, there is a literature gap on the effect of ozonation on the chemical composition of fish as well as other determinants of fish quality.”

Language and Clarity Issues:

The manuscript requires extensive language revision to improve clarity and readability. Many sections contain ambiguous statements, grammar errors, and redundancies, which hinder the reader’s ability to understand the study's findings and implications.

Action taken:

Manuscript language was improved throughout the manuscript.

Minor Concerns

Figures and Tables:

Some figures, such as those showing the malondialdehyde levels and microbial load, are not labeled clearly, and it is difficult to interpret their significance without adequate legends and discussion.

Action taken:

The figures quality and their description were improved.

Table 2, which presents the volatile fraction, lacks proper labeling of columns and rows, making it challenging to understand the changes in volatile compounds.

Action taken: Table 2 was improved.

Reviewer #2: This manuscript is about the Impact of gaseous ozone treatment of fish carcasses (Gadus morhua) on the microbiological load and their quality. My recommendation on submitting the manuscript to PLOS ONE is acceptation after major revision. Please see detailed comments and questions as below

-In Title, Scientific name (Gadus morhua) should be written in italic format.

Action taken: done

-In the abstract, it is better to mention the values of the results obtained.

Action taken: abstract section was improved by the addition of microbial load values

For example, for the control sample with 1.8 x 103 cfu, which was exposed to 20 ppm ozone in atmosphere during the exposition time in the range 10-30 min, a microbial load was reduced to a level between 1.6 x 103 and 1.2 x 103 cfu. The observed reduction levels indicated that such an ozone treatment procedure can be declared as an viable option for improvement of microbial safety of fish

- At the end of abstract, the keywords starts with first capital letter.

Action taken: keywords were corrected

-The introduction section is long and needs to be rewritten.

Action taken: the introduction section was significantly shortened.

-Due to the effect of ozone on food fat and the creation of an undesirable taste, It is better that sensory evaluation performed on the samples.

Action taken:

In section “Sensory analysis” a proper comment was added:

A preliminary sensory analysis of the fish tissue performed without heat treatment indicated the presence of foreign odour notes, which are typical of carbonyl products of fatty acid decomposition. Furthermore, in the volatile fraction, a representative of these compounds was identified, i.e., reported above pentadecanal (see Table 2). Due to its high volatility, this compound will not have a significant impact on the perception of consumers of heat treated products.

-Comparison of results with those of other similar studies, as well as discussion in the results section, is weak.

Response:

There is a literature gap on the subject of complex effects of ozonation treatment. In particular, only microbial load effect is investigated, while effects like changes in fatty acid profile or volatile faction composition were not the subject of any of known literature positions available.

Action taken:

In the introduction section a proper comment was added:

“Although there is a great deal of data on ozone treatment in regard to microbial safety, there is a lack of data on the complex effects of the interaction of ozone with fish products. In particular, there is a literature gap on the effect of ozonation on the chemical composition of fish as well as other determinants of fish quality.”

-Please insert significant letters above the columns in figures.

Action taken: significant letters were added in figures

---

## [Decision Letter · Decision Letter 1]

2 Apr 2025

PONE-D-24-48297R1Impact of gaseous ozone treatment of fish carcasses (Gadus morhua) on the microbiological load and their quality.

PLOS ONE

Dear Dr. Antos,

Thank you for submitting your manuscript to PLOS ONE. After careful consideration, we feel that it has merit but does not fully meet PLOS ONE’s publication criteria as it currently stands. Therefore, we invite you to submit a revised version of the manuscript that addresses the points raised during the review process.

======

Note from the Editorial Office: We note that you have included the results of a new sensory analysis in your manuscript, but have not reported the details of how it was performed in your Methods section. Please include a detailed description of the sensory analysis in your Methods.

Additionally, if your sensory analysis involved human participants, PLOS One requires authors to confirm that this specific study was reviewed and approved by an institutional review board (ethics committee) before the study began. Please provide the specific name of the ethics committee/IRB that approved your study, or explain why you did not seek approval in this case. Once you have amended this/these statement(s) in the Methods section of the manuscript, please add the same text to the “Ethics Statement” field of the submission form (via “Edit Submission”).

======

We look forward to receiving your revised manuscript.

Kind regards,

Sarah Jose

Staff Editor

PLOS ONE

On behalf of:

José M. Alvarez-Suarez

Academic Editor

PLOS ONE

Journal Requirements:

Additional Editor Comments (if provided):

Reviewers' comments:

Reviewer's Responses to Questions

**Comments to the Author**

1. If the authors have adequately addressed your comments raised in a previous round of review and you feel that this manuscript is now acceptable for publication, you may indicate that here to bypass the “Comments to the Author” section, enter your conflict of interest statement in the “Confidential to Editor” section, and submit your "Accept" recommendation.

Reviewer #1: All comments have been addressed

Reviewer #2: All comments have been addressed

Reviewer #3: All comments have been addressed

2. Is the manuscript technically sound, and do the data support the conclusions?

Reviewer #1: No

Reviewer #2: Yes

Reviewer #3: Yes

3. Has the statistical analysis been performed appropriately and rigorously? 

Reviewer #1: No

Reviewer #2: Yes

Reviewer #3: Yes

4. Have the authors made all data underlying the findings in their manuscript fully available?

Reviewer #1: Yes

Reviewer #2: Yes

Reviewer #3: Yes

5. Is the manuscript presented in an intelligible fashion and written in standard English?

Reviewer #1: Yes

Reviewer #2: Yes

Reviewer #3: Yes

6. Review Comments to the Author

Reviewer #1: Author responses are not satisfactory. Also, sensory analysis requires an ethical permission. Table 2 is too complex to understand.

Hence, I have to reject.

Reviewer #2: (No Response)

Reviewer #3: (No Response)

7. PLOS authors have the option to publish the peer review history of their article (what does this mean? ). If published, this will include your full peer review and any attached files.

**Do you want your identity to be public for this peer review?** For information about this choice, including consent withdrawal, please see our Privacy Policy .

Reviewer #1: No

Reviewer #2: **Yes: ** Hassan Barzegar

Reviewer #3: **Yes: ** Ahmed Noah Badr

---

## [Author Response · Author response to Decision Letter 2]

3 May 2025

Note from Editor:

Note from the Editorial Office: We note that you have included the results of a new sensory analysis in your manuscript, but have not reported the details of how it was performed in your Methods section. Please include a detailed description of the sensory analysis in your Methods.

Additionally, if your sensory analysis involved human participants, PLOS One requires authors to confirm that this specific study was reviewed and approved by an institutional review board (ethics committee) before the study began. Please provide the specific name of the ethics committee/IRB that approved your study, or explain why you did not seek approval in this case. Once you have amended this/these statement(s) in the Methods section of the manuscript, please add the same text to the “Ethics Statement” field of the submission form (via “Edit Submission”).

Action taken

The description of sensory analysis has been improved in the methods section:

Method of sensory analysis

As a part of the conducted research, the chemical investigations were supplemented only by a preliminary sensory analysis. For this purpose, the samples that were intended to be subjected to the SPME analysis were previously subjected to a preliminary sensory analysis by the authors of the manuscript, who described the sensory experiences while smelling the sample. The authors were trained to be able to participate in sensory analysis of food products and therefore also ozonated fish matrix. All of them volunteered as the subjects for sensory analysis of fish matrix exposed to ozonation. Therefore, no additional approval of ethic committee was asked for.

The discussion on sensory analysis was updated:

Discussion on sensory analysis

Finally a preliminary sensory analysis of the fish tissue performed without heat treatment indicated the presence of foreign odour notes, which are typical of carbonyl products of fatty acid decomposition. Furthermore, in the volatile fraction, a representative of these compounds was identified, i.e., reported above pentadecanal (see section above: Analysis of volatile faction, Table 2). Due to its high volatility, this compound will not have a significant impact on the perception of consumers of heat treated products.

---

## [Decision Letter · Decision Letter 2]

30 May 2025

PONE-D-24-48297R2Impact of gaseous ozone treatment of fish carcasses (Gadus morhua) on the microbiological load and their quality.PLOS ONE

Dear Dr. Antos,

Thank you for submitting your manuscript to PLOS ONE. After careful consideration, we feel that it has merit but does not fully meet PLOS ONE’s publication criteria as it currently stands. Therefore, we invite you to submit a revised version of the manuscript that addresses the points raised during the review process.

Request from the Editorial Office:

In line with the principles expressed in the Declaration of Helsinki, we expect all research involving human participants and/or medical data to have been approved by the authors' Institutional Review Board (IRB) or by equivalent ethics committee(s). If the need for ethical approval is waived, this should be formally confirmed by a suitable committee generally before the start of the study.

As the requirements of journals and institutions are becoming stricter, approval from an independent ethics committee is becoming the norm for all research studies involving human participants and/or medical information independently of how low the risks are. Please note that we reserve the right to reject any submission that does not meet these standards, which in some cases are more stringent than local ethical standards.

Therefore before we can proceed further, we would require your institutional ethics committee to formally confirm that ethical approval was not needed in this case. Please note that we do not accept retrospective ethics approval. (If approval is needed, this should have been done before the start of the study.) Please include a copy of the letter from the ethics committee as an "Other" file.

We hope you understand the reasons behind this request and look forward to hearing from you.

We look forward to receiving your revised manuscript.

Kind regards,

José M. Alvarez-Suarez

Academic Editor

PLOS ONE

Journal Requirements:

Reviewers' comments:

Reviewer's Responses to Questions

**Comments to the Author**

1. If the authors have adequately addressed your comments raised in a previous round of review and you feel that this manuscript is now acceptable for publication, you may indicate that here to bypass the “Comments to the Author” section, enter your conflict of interest statement in the “Confidential to Editor” section, and submit your "Accept" recommendation.

Reviewer #2: All comments have been addressed

Reviewer #4: All comments have been addressed

2. Is the manuscript technically sound, and do the data support the conclusions?

Reviewer #2: Yes

Reviewer #4: Yes

3. Has the statistical analysis been performed appropriately and rigorously? 

Reviewer #2: Yes

Reviewer #4: Yes

4. Have the authors made all data underlying the findings in their manuscript fully available?

Reviewer #2: Yes

Reviewer #4: Yes

5. Is the manuscript presented in an intelligible fashion and written in standard English?

Reviewer #2: Yes

Reviewer #4: Yes

6. Review Comments to the Author

Reviewer #2: (No Response)

Reviewer #4: Dear author/s

I just evaluated the paper in revise format and check the answers to all the questions. I think the paper reponse all the questions and can be accepted for publication.

Best Regards

7. PLOS authors have the option to publish the peer review history of their article (what does this mean? ). If published, this will include your full peer review and any attached files.

**Do you want your identity to be public for this peer review?** For information about this choice, including consent withdrawal, please see our Privacy Policy .

Reviewer #2: **Yes: ** Hassan Barzegar

Reviewer #4: No

---

## [Author Response · Author response to Decision Letter 3]

17 Jun 2025

Answer for the Editor:

The authors have decided to remove the section describing human subjects research from the manuscript. We hope the revised version will still be considered suitable for publication

Action taken:

The description of sensory analysis and its discussion have been removed from the manuscript.

---

## [Editor Report · Decision Letter 3]

24 Jun 2025

Impact of gaseous ozone treatment of fish carcasses (Gadus morhua) on the microbiological load and their quality.

PONE-D-24-48297R3

Dear Dr. Antos,

We’re pleased to inform you that your manuscript has been judged scientifically suitable for publication and will be formally accepted for publication once it meets all outstanding technical requirements.

Kind regards,

José M. Alvarez-Suarez

Academic Editor

PLOS ONE
---

## [Editor Report · Acceptance letter]

PONE-D-24-48297R3

PLOS ONE

Dear Dr. Antos,

I'm pleased to inform you that your manuscript has been deemed suitable for publication in PLOS ONE. Congratulations! Your manuscript is now being handed over to our production team.

Kind regards,

on behalf of

Professor José M. Alvarez-Suarez

Academic Editor

PLOS ONE